# Alphabetical ordering of author surnames in academic publishing: A detriment to teamwork

Steven T. Joanis[1]*, Vivek H. Patil[2]

1 Heider College of Business, Creighton University, Omaha, Nebraska, United States of America, 2 School of Business Administration, Gonzaga University, Spokane, Washington, United States of America

* stj95568@creighton.edu

## Abstract

**Data Availability Statement:** We uploaded all data required to replicate all analyses conducted in the writing of this manuscript. The data depository used is: DANS (Data Archiving and Networked

### Introduction

In academia, many institutions use journal article publication productivity for making decisions on tenure and promotion, funding grants, and rewarding stellar scholars. Although non-alphabetical sequencing of article coauthoring by the spelling of surnames signals the extent to which a scholar has contributed to a project, many disciplines in academia follow the norm of alphabetical ordering of coauthors in journal publications. By assessing business academic publications, this study investigates the hypothesis that author alphabetical ordering disincentivizes teamwork and reduces the overall quality of scholarship.

### Methods

To address our objectives, we accessed data from 21,353 articles published over a 20-year period across the four main business subdisciplines. The articles selected are all those published by the four highest-ranked journals (in each year) and four lower-ranked journals (in each year) for accounting, business technology, marketing, and organizational behavior. Poisson regression and binary logistic regression were utilized for hypothesis testing.

### Results

This study finds that, although team size among business scholars is increasing over time, alphabetical ordering as a convention in journal article publishing disincentivizes author teamwork. This disincentive results in fewer authors per publication than for publications using contribution-based ordering of authors. Importantly, article authoring teamwork is related to article quality. Specifically, articles written by a single author typically are of lesser quality than articles published by coauthors, but the number of coauthors exhibits decreasing returns to scale—coauthoring teams of one to three are positively related to high-quality articles, but larger teams are not. Alphabetical ordering itself, however, is positively associated with quality even though it inhibits teamwork, but journal article coauthoring has a greater impact on article quality than does alphabetical ordering.

Services). The DOI is: https://doi.org/10.17026/dans-z72-4qp3.

**Funding:** The authors received no specific funding for this work.

**Competing interests:** The authors have declared that no competing interests exist.

## Conclusions

These findings have important implications for academia. Scholars respond to incentives, yet alphabetical ordering of journal article authors conflicts with what is beneficial for the progress of academic disciplines. Based on these findings, we recommend that, to drive the highest-quality research, teamwork should be incentivized—all fields should adopt a contribution-based journal article author-ordering convention and avoid author ordering based upon the spelling of surnames. Although this study was undertaken using articles from business journals, its findings should generalize across all academia.

## Introduction

### Theoretical background

Academic scholars respond to incentives, and their responses could be in conflict with what is beneficial for the progress of academic disciplines [1, 2]. One major source of recognition for scholars is the publication of their work in academic journals. When collaborating, academics make the important decision of determining the sequence of authorship attribution [3]. In certain disciplines—such as mathematics and economics—listing coauthors alphabetically by surname generally is the norm. In many other disciplines—such as psychology and sociology—the coauthorship sequence is determined based on the level of contribution by coauthors [4–6]. In disciplines that use contribution-based authorship determination, the first-listed author position is considered important because it indicates that the first-listed author's contributions were more substantial than those of the coauthors.

Why is it important to understand how coauthorship is determined? Primarily because the productivity of scholars is used by many institutions—including academic departments, schools, and universities—for making decisions on tenure and promotion, funding grants, and rewarding stellar scholars. The authorship sequence for articles potentially signals the extent to which a scholar has contributed to a project. This signal is clearer when authorship is determined based on the author's contributions to a project. The signal is ambiguous when alphabetical order is used because such ordering hides the relative contributions of individual authors [7]. As a result, alphabetical ordering provides an advantage to authors whose surnames begin with letters falling at the beginning of the alphabet. For authors whose surnames start with letters in the middle or toward the end of the alphabet, however, use of alphabetical order creates a disincentive to collaborate with coauthors whose surnames would precede theirs alphabetically (e.g., [3, 6, 8]). Indeed, this disincentive has led some authors with surnames beginning with letters that fall at the end of the alphabet to collaborate less frequently and to instead publish under their own name as the sole author [3]. This result is detrimental to disciplines, as collaborations have been found to lead to higher-quality articles [9]. Thus, understanding authorship-assignment practices is important because they can affect careers and impact the quality of work conducted within a discipline [8, 10].

The main purpose of this study is to advance the literature by assessing author-ordering conventions and the relationship between these ordering conventions and the number of coauthors of articles and the quality of articles. We also examine how these relationships have changed over time. Specifically, we focus here on the business academic literature, where our review indicates that there is no study that has investigated these issues.

## Hypotheses development

The business literature is not dissimilar to other disciplines—when collaborators publish their findings, they list coauthor names either alphabetically by surname or based on coauthor contribution level [3]. Furthermore, certain disciplines (such as accounting) follow a general convention of listing coauthors alphabetically, but many others (such as marketing) use contribution-based sequencing [5, 6, 11].

Collaboration among scholars is on the rise, as evidenced by the number of coauthored articles in academic publications [12]. This increase in the size of author teams is seen across nearly all academic disciplines that have been previously studied [6, 13]. Additionally, the research of Kendall and colleagues [14] indicates that millennials are more likely to embrace teamwork than are the generations that preceded them, suggesting that teamwork should increase as millennials become a greater percentage of academia. However, this effect has not been studied in the business academic research, motivating our first hypothesis: *Hypothesis 1*: *Team size among business scholars is increasing over time.*

Given the disincentive for scholars with surnames that begin with letters toward the end of the alphabet, alphabetical ordering of authorship is likely to inhibit teamwork (e.g., [3, 6, 8]). In contrast, contribution-based authorship sequencing is likely to encourage collaboration between a greater number of coauthors. This view is supported by Maciejovsky and colleagues [5], who state that, "ordering authors by relative contributions provides incentives to include more authors as a research project progresses and allows rewarding those authors according to a smaller cost to oneself than the case of alphabetical ordering" [5, p. 597]. Thus, our second hypothesis is stated as follows: *Hypothesis 2*: *Alphabetical ordering disincentivizes author teamwork and results in fewer authors per publication than does contribution-based ordering of authors.*

Academic disciplines—which are simply a collection of individual academics—have the goals of advancing their discipline and fostering scholarship [15]. Achieving these goals should maximize the utility of the individuals that comprise an academic discipline. A key driver of success is teamwork [16–19], which should lead to higher-quality work. Prior literature has suggested that the quality of articles—as reflected by the ranking of journals in which they are published—is related to author ordering conventions (e.g., [7, 20, 21]). The contention this stream of literature makes is that the greater stringency with which papers are accepted in higher-ranked publication outlets forces coauthors to perform at their best. As a result, the contribution levels are approximately similar and coauthors therefore choose to list themselves alphabetically. We concede the possibility that equivalent contributions could lead coauthors to list their names alphabetically. However, we subscribe to Wuchty and colleagues' finding [13] that teamwork—as reflected by team size—is critical to producing high-quality work. This motivates our next two hypotheses: *Hypothesis 3*: *Teamwork is related to article quality.* *Hypothesis 4*: *Journal article coauthoring has a greater impact on article quality than does alphabetical ordering.*

## Methods

### Data

This study focuses on business academia. Based upon the hypotheses to be assessed, the dataset required journal ranking information. Business academic journal rankings, like the hard sciences (e.g., chemistry or physics) and social sciences (e.g., psychology or economics), is sub-discipline based, requiring us to identify high and lower ranked journals based upon subdisciplines. The subdisciplines of accounting, organizational behavior, business technology, and

marketing were selected for assessment simply because they are subdisciplines that are represented in nearly every accredited business school worldwide. For each of these subdisciplines and each year between 1999 and 2018, we identified eight journals that were classified as belonging to either the "Top Journal" or the "Other Journal" categories (defined further below) using existing rankings from the SCImago database. Much of the literature on publication rankings uses either SCImago Journal Rank (SJR) or Thomson Reuters' Journal Impact Factor (JIF). We use SJR because it takes into consideration not only citation numbers but also journal prestige (as opposed to only popularity and impact, as is the case for JIF) [22]. Guerrero-Bote and Moya-Anegon [23] found that the SJR rankings—even with their greater weighting of prestige as compared to JIF—still had strongly correlated rankings ($r = 0.944$), meaning that our choice of ranking source likely would have little impact on study results. We define the Top Journals in each subdiscipline as those ranked between 1 and 4 in each year. To create separation in quality between the Top Journals and those not considered top journals, Other Journals are those that were ranked 11 through 14 in every comparison year.

After identifying journals for each subdiscipline for each year, we used the Scopus database (the major source for SJR) to identify all articles published in those journals. In some cases, SJR-ranked journals did not have any content identified as traditional journal articles in a given year (i.e., all articles were identified as surveys, editorials, or some other non-article type). In that case, it was replaced with the next-highest ranked journal in SJR. This resulted is a dataset composed of 31,512 records; 27.5% of these were in accounting, 27.3% in business technology, 25.2% in marketing, and 20.0% in organizational behavior. The sample size was further reduced by removing articles that were not research articles (e.g., editorials) and articles with more than eight coauthors (as they were outliers), resulting in a dataset of 26,720 articles. Lastly, we removed all single-author articles, which resulted in a sample of 21,362 articles.

**Variables.** The variable of interest in this study is the number of coauthors of a journal article—a proxy for teamwork—which was directly identified for each journal article. Author ordering is a categorical variable that identifies whether the authors of a journal article were alphabetically ordered. Articles were also categorized as high ranked (from journals ranked 1 to 4) and lower ranked (from journals ranked 11 to 14). The article publication years range from 1999 to 2018, as noted above.

## Statistical analyses

In addition to the analysis of summary statistics and the assessment of univariate Pearson correlations, we used three main analyses for hypotheses testing. The first analysis is employed to assess Hypothesis 1 and Hypothesis 2. As a reminder, Hypothesis 1 suggests that team size among business researchers is increasing over time, and Hypothesis 2 posits that alphabetical ordering disincentivizes author teamwork and results in fewer authors per publication than contribution-based non-alphabetical ordering of authors. The dependent variable here is the number of coauthors, but it is transformed by netting 1 from each integer value. The range is therefore 0 to 6 and allows the use of Poisson regression—the proper analysis for bounded integer-dependent variables. The independent variables in this analysis are the alphabetical ordering categorical variable and year fixed effects.

The second analysis, used to assess Hypothesis 3 (that research team size is positively related to article quality), is conducted using binary logistic regression because the dependent variable is either an article from a top-rated journal or is not and is predicted by either the number of coauthors or a set of categorical variables indicating the number of coauthors (e.g., one author, two coauthors). Again, year fixed effects are used as a control variable because this effect could be changing over time.

The final analysis is also a binary logistic regression and is used to assess Hypothesis 4, that journal article coauthoring has a greater impact on article quality than does alphabetical ordering. Here, article quality is predicted by alphabetical ordering and a new variable that indicates whether the article is authored by a single person or a team, controlling for year fixed effects. Additionally, the dataset for this final analysis is slightly different than the dataset used for the previous analyses in that this dataset includes articles that have a sole author, whereas the previous analyses excluded single-author articles. The sample size increased from 21,362 to 26,720–20.1% of the articles are by sole authors. All analyses were performed using R version 3.6.1 [24].

## Results

### Descriptive statistics

Table 1 shows summary statistics for the dataset.

The 61 publications with more than 7 coauthors were considered outliers and removed from the sample. The resultant sample ranges from 1 to 7 coauthors, with a mean number of coauthors of 1.71, and a distribution as shown in Fig 1. Of the total, 56.0% of the journal articles list authors in alphabetical order; however, even for business subdisciplines in which the convention is to order authors by contribution (e.g., marketing), random chance would dictate that some articles still would be in alphabetical order. Of the total articles, 57.8% comes from high-ranked journals (indicating that higher-ranked journals have more articles per year), and the breakdown of articles by discipline is 28.4% accounting, 26.8% business technology, 26.0% marketing, and 18.8% organizational behavior. Finally, there appears to be a general increase in the number of articles per journal over time.

Bivariate correlations are shown in Table 2.

Table 2 shows that the number of article coauthors is significantly and negatively bivariately correlated with alphabetical ordering of authors ($r(21,362) = -0.307$, $p < .001$), giving initial credence to Hypothesis 2, namely that alphabetical ordering impedes teamwork, with the number of coauthors as the proxy for teamwork. This analysis also shows that higher-ranked journals have fewer coauthors ($r(21,362) = -0.026$, $p < .001$) and tend to favor alphabetical listing of authors ($r(21,362) = 0.114$, $p < .001$). Although this is a preliminary analysis, this is the opposite of what is suggested by Hypothesis 4, which hypothesizes that journal article coauthoring would have a *greater* impact on article quality than alphabetical ordering would. Here, however, the magnitude of the correlation between high rank and alphabetical ordering is more than four times that of the correlation between high rank and the number of coauthors.

Table 2 also demonstrates the phenomena discussed above, that certain business subdisciplines have alphabetical author ordering conventions while other subdisciplines do not. Accounting is strongly and statistically positively correlated with alphabetical ordering, indicating that the convention in that field is to order authors alphabetically ($r(21,362) = 0.454$, $p < .001$). The other three fields—organizational behavior, business technology, and marketing—anecdotally follow a contribution-based author-ordering convention, demonstrated by negative and statistically significant correlations with alphabetical ordering (organizational behavior: $r(21,362) = -0.129$, $p < .001$; business technology: $r(21,362) = -0.150$, $p < .001$; marketing: $r(21,362) = -0.200$, $p < .001$).

### Generalized model equation

There are several models employed for hypothesis testing. For Hypothesis 1 and Hypothesis 2, the number of coauthors is predicted by alphabetical ordering, controlling for year fixed

**Table 1. Number of coauthors of articles published by year.**

| Statistic | N | Mean | St. Dev. | Min. | Max. |
|---|---|---|---|---|---|
| Number of Coauthors | 21,362 | 1.711 | 0.847 | 1 | 7 |
| 1 Coauthor | 21,362 | 0.480 | 0.500 | 0 | 1 |
| 2 Coauthors | 21,362 | 0.378 | 0.485 | 0 | 1 |
| 3 Coauthors | 21,362 | 0.108 | 0.310 | 0 | 1 |
| 4 Coauthors | 21,362 | 0.025 | 0.155 | 0 | 1 |
| 5 Coauthors | 21,362 | 0.007 | 0.085 | 0 | 1 |
| 6 Coauthors | 21,362 | 0.002 | 0.046 | 0 | 1 |
| 7 Coauthors | 21,362 | 0.001 | 0.024 | 0 | 1 |
| Alphabetical Order | 21,362 | 0.560 | 0.496 | 0 | 1 |
| Non-Alphabetical Order | 21,362 | 0.440 | 0.496 | 0 | 1 |
| Journal Ranking: | | | | | |
| High Rank = 1 | 21,362 | 0.578 | 0.494 | 0 | 1 |
| Low Rank = 1 | 21,362 | 0.422 | 0.494 | 0 | 1 |
| Publication Year: | | | | | |
| 1999 | 21,362 | 0.035 | 0.183 | 0 | 1 |
| 2000 | 21,362 | 0.042 | 0.200 | 0 | 1 |
| 2001 | 21,362 | 0.033 | 0.178 | 0 | 1 |
| 2002 | 21,362 | 0.033 | 0.178 | 0 | 1 |
| 2003 | 21,362 | 0.033 | 0.180 | 0 | 1 |
| 2004 | 21,362 | 0.033 | 0.179 | 0 | 1 |
| 2005 | 21,362 | 0.037 | 0.190 | 0 | 1 |
| 2006 | 21,362 | 0.042 | 0.201 | 0 | 1 |
| 2007 | 21,362 | 0.048 | 0.215 | 0 | 1 |
| 2008 | 21,362 | 0.050 | 0.218 | 0 | 1 |
| 2009 | 21,362 | 0.057 | 0.232 | 0 | 1 |
| 2010 | 21,362 | 0.066 | 0.248 | 0 | 1 |
| 2011 | 21,362 | 0.071 | 0.257 | 0 | 1 |
| 2012 | 21,362 | 0.056 | 0.230 | 0 | 1 |
| 2013 | 21,362 | 0.067 | 0.250 | 0 | 1 |
| 2014 | 21,362 | 0.064 | 0.245 | 0 | 1 |
| 2015 | 21,362 | 0.054 | 0.227 | 0 | 1 |
| 2016 | 21,362 | 0.062 | 0.241 | 0 | 1 |
| 2017 | 21,362 | 0.062 | 0.240 | 0 | 1 |
| 2018 | 21,362 | 0.055 | 0.228 | 0 | 1 |

effects. The modelling Eq (1) is shown below.

$$\text{Number of Coauthors} = B_0 +$$
$$B_1 * \text{If Alphabetically Ordered} +$$
$$B_2 * \text{Year Fixed Effects Matrix} +$$
$$e \tag{1}$$

Alphabetical ordering is a categorical variable, and 1 is used in the case where alphabetical ordering is exhibited, 0 is used when it is not. The number of coauthors is an integer bounded at 1 with a maximum of 7 coauthors. Year fixed effects also are categorical, representing the year in which an article was published and ranging from 1999 through 2018.

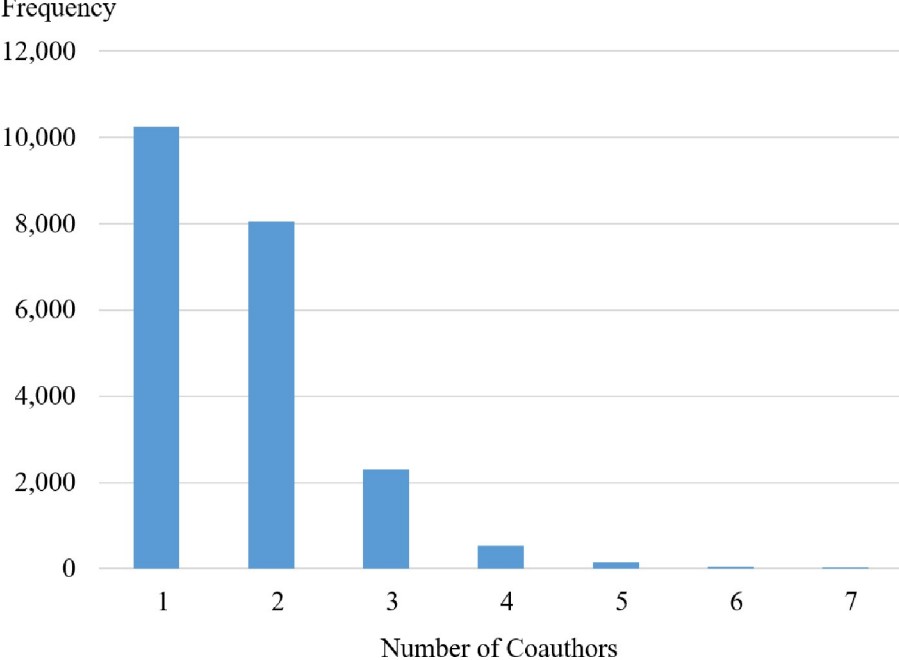

**Fig 1. Number of coauthors.**

The model for Hypothesis 3 predicts article quality as a function of the number of authors, again controlling for year fixed effects, and is shown as Eq (2) below.

$$
\begin{aligned}
\text{Article Quality} = \quad & B_0+ \\
& B_1 * \text{Number of Coauthors}+ \\
& B_2 * \text{Number of Coauthor Categorical Matrix}+ \\
& B_3 * \text{Year Fixed Effects Matrix}+ \\
& e
\end{aligned}
\tag{2}
$$

In this case, article quality is a binary variable with highly ranked articles categorized as a 1 and other articles categorized as a 0. The number of coauthors is as described above, and is an

**Table 2. Pearson correlation matrix.**

| Variable | Number of Coauthors | Alpha. | Not Alpha. | High Rank | Low Rank | Acct. | Org. Behavior | Bus. Tech. |
|---|---|---|---|---|---|---|---|---|
| **Alpha.** | -0.307*** | 1.000*** | | | | | | |
| **Not Alpha.** | 0.307*** | -1.000*** | 1.000*** | | | | | |
| **High Rank** | -0.026*** | 0.114*** | -0.114*** | 1.000*** | | | | |
| **Low Rank** | 0.026*** | -0.114*** | 0.114*** | -1.000*** | 1.000*** | | | |
| **Accounting** | -0.067*** | 0.454*** | -0.454*** | 0.150*** | -0.150*** | 1.000*** | | |
| **Org. Behavior** | 0.056*** | -0.129*** | 0.129*** | -0.101*** | 0.101*** | -0.303*** | 1.000*** | |
| **Bus. Tech.** | 0.015* | -0.150*** | 0.150*** | -0.067*** | 0.067*** | -0.381*** | -0.291*** | 1.000*** |
| **Marketing** | 0.003 | -0.200*** | 0.200*** | 0.004 | -0.004 | -0.373*** | -0.285*** | -0.359*** |

*$p < .050$

**$p < .010$

***$p < .001$; $p$-values are reported based upon $t$-stats.

integer ranging from 0 to 7 (with the regression coefficient represented by $B_1$ in Eq 2). In some treatments of the model, however, instead of the integer number of coauthors, a matrix of categorical variables is used to represent each integer value of the coauthor predictor variable (with the regression coefficient represented as $B_2$ in the equation). In no treatments of the model are both of these predictor variables used simultaneously, as this would cause singularity in the regression.

Our final model Eq (3) is employed primarily to assess Hypothesis 4, but also to further assess Hypothesis 3. This analysis predicts article quality as a function of alphabetical ordering and whether an article is published by an individual or a team (controlling for year fixed effects).

$$
\begin{aligned}
\text{Article Quality} = \quad & B_0 + \\
& B_1 * \text{If Alphabetically Ordered} + \\
& B_2 * \text{If Article is Coauthored} + \qquad (3) \\
& B_3 * \text{Year Fixed Effects Matrix} + \\
& e
\end{aligned}
$$

All variables in this model are as described above, with the exception of whether an article is coauthored—this model uses a dataset that includes single-author papers, thus the coauthored variable here is a binary variable with a value of 0 if sole-authored and a value of 1 if authored by a team of researchers.

## Hypothesis testing

Our first analysis is used to assess both Hypothesis 1 and Hypothesis 2. Hypothesis 1 posits that team size among business scholars is increasing over time. Hypothesis 2 suggests that alphabetical ordering disincentivizes author teamwork and will result in fewer authors per publication than contribution-based ordering of authors. To assess these hypotheses, we predict the number of coauthors by whether the publication is alphabetically ordered and by year of publication. As discussed above, to meet the assumptions of the required Poisson regression, the coauthor variable is transformed by subtracting 1 from each value. Our findings are summarized in Table 3.

For both treatment (1) and treatment (2), the year coefficients are statistically significant for each year, indicating that they are all different from the analysis null, which is the year 2018. By graphing the value of these coefficients (Fig 2), we see an increase in their value over time. These findings indicate support for Hypothesis 1, that team size among business scholars is increasing over time.

The negative and statistically significant coefficient for alphabetical ordering of authors in treatment (2) ($B_1$ = -0.731, $p < .001$) indicates that articles that are alphabetically ordered have fewer authors. This finding indicates support for Hypothesis 2. Alphabetical ordering disincentivizes author teamwork, resulting in fewer authors per publication than for contribution-based ordering of authors.

The next analysis assesses Hypothesis 3, that article authoring team size is related to article quality. To assess this hypothesis, we predict article quality by alphabetical ordering and several measures of team size to compare the impact of these predictors on article quality. Our findings are summarized in Table 4.

Treatment (1) predicts article quality by the number of authors, controlling only for time fixed effects. The negative and statistically significant coefficient ($B_1$ = -0.062, $p < .001$) indicates that teamwork is detrimental to quality. To gain a greater degree of fidelity in the analysis,

**Table 3. Poisson regression: Dependent variable is number of article coauthors minus 1.**

| | (1) | (2) |
|---|---|---|
| **Constant** | -0.045 | 0.298*** |
| | (-1.490) | (9.743) |
| **Alpha. Ordered** | | -0.731*** |
| | | (-43.674) |
| **Time (2018 = 0)** | | |
| 1999 | -0.664*** | -0.622*** |
| | (-10.989) | (-10.294) |
| 2000 | -0.614*** | -0.591*** |
| | (-11.094) | (-10.665) |
| 2001 | -0.643*** | -0.603*** |
| | (-10.510) | (-9.861) |
| 2002 | -0.456*** | -0.441*** |
| | (-7.980) | (-7.716) |
| 2003 | -0.508*** | -0.514*** |
| | (-8.805) | (-8.907) |
| 2004 | -0.528*** | -0.525*** |
| | (-9.076) | (-9.025) |
| 2005 | -0.511*** | -0.508*** |
| | (-9.221) | (-9.168) |
| 2006 | -0.358*** | -0.379*** |
| | (-7.091) | (-7.500) |
| 2007 | -0.402*** | -0.396*** |
| | (-8.195) | (-8.078) |
| 2008 | -0.277*** | -0.278*** |
| | (-5.930) | (-5.960) |
| 2009 | -0.398*** | -0.371*** |
| | (-8.551) | (-7.971) |
| 2010 | -0.307*** | -0.317*** |
| | (-7.041) | (-7.255) |
| 2011 | -0.276*** | -0.307*** |
| | (-6.501) | (-7.236) |
| 2012 | -0.199*** | -0.224*** |
| | (-4.495) | (-5.074) |
| 2013 | -0.208*** | -0.207*** |
| | (-4.915) | (-4.890) |
| 2014 | -0.193*** | -0.183*** |
| | (-4.516) | (-4.284) |
| 2015 | -0.167*** | -0.160*** |
| | (-3.770) | (-3.611) |
| 2016 | -0.132** | -0.146*** |
| | (-3.122) | (-3.436) |
| 2017 | -0.125** | -0.123** |
| | (-2.953) | (-2.898) |
| **Observations** | 21,362 | 21,362 |
| **Log Likelihood** | -23,485.1 | -22,494.7 |

*(Continued)*

**Table 3.** (Continued)

| | (1) | (2) |
|---|---|---|
| **Akaike Inf. Crit.** | 47,010.3 | 45,031.3 |

$^{*}p < .050$

$^{**}p < .010$

$^{***}p < .001$; $z$-stats are reported parenthetically below each coefficient.

treatment (2) uses categorical variables to indicate smaller teams, with indicators for sole authors and teams of 2 or 3 coauthors, and treatment (3) uses categorical variables to indicate larger teams of 4, 5, or 6 coauthors. As indicated by the positive coefficients for the smaller number of coauthors in treatment (2), we find that teams of 3 or fewer coauthors are associated with higher-quality journals (1 author: ($B_1 = 0.486$, $p < .001$); 2 coauthors: ($B_1 = 0.571$, $p < .001$); 3 coauthors: ($B_1 = 0.437$, $p < .001$). However, the negative coefficients for the number of authors in treatment (3) indicate that coauthors teams of 4 or more generally are associated with lower-quality journals (4 coauthors: ($B_1 = -0.479$, $p < .001$); 5 coauthors: ($B_1 = -0.726$, $p < .001$). For 6 coauthors, the coefficient is not significant ($B_1 = 0.437$, $p < .106$), so no conclusions can be drawn. These findings are consistent with treatment (1). Although this might seem counterintuitive, this effect is driven by alphabetical ordering which, as shown, inhibits teamwork. This effect is highlighted by the dampening effect on all the coauthor categorical coefficients by the inclusion of the alphabetical coefficient in treatment (5) and treatment (6) (e.g., all the categorical coefficients are closer to zero when including the alphabetical indicator than when it is excluded). Hypothesis 3 is supported. Article authoring team size is related to article quality. Specifically, coauthoring teams of three or fewer are positively related to high-quality articles, and larger teams are not.

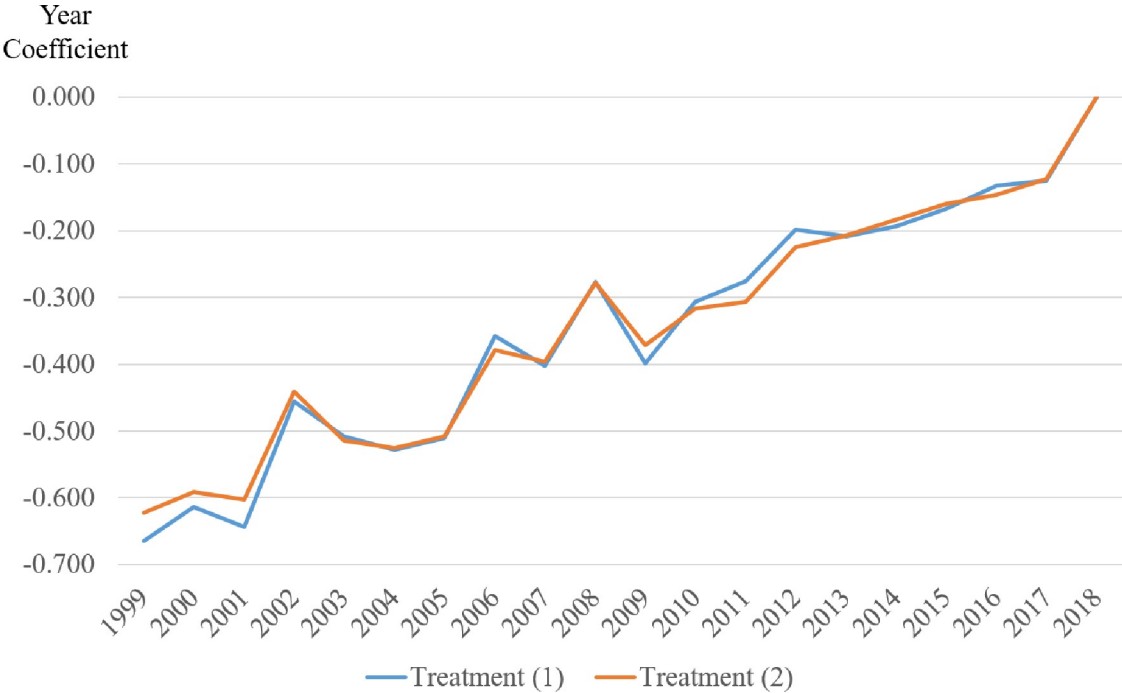

**Fig 2. Trend of time coefficient as a predictor of the number of article coauthors.**

Table 4. Binary logistic regression: Dependent variable is article quality (High = 1).

| | (1) | (2) | (3) | (4) | (5) |
|---|---|---|---|---|---|
| **Constant** | 0.439*** | -0.226* | 0.283*** | -0.300*** | 0.025 |
| | (5.729) | (-2.448) | (4.786) | (-3.224) | (0.405) |
| **Number of Authors** | -0.062*** | | | | |
| | (-3.744) | | | | |
| **Alpha Ordering** | | | | 0.485*** | 0.447*** |
| | | | | (16.348) | (15.651) |
| **1 Coauthor** | | 0.486*** | | 0.202* | |
| | | (6.313) | | (2.557) | |
| **2 Coauthors** | | 0.571*** | | 0.386*** | |
| | | (7.356) | | (4.911) | |
| **3 Coauthors** | | 0.437*** | | 0.340*** | |
| | | (5.131) | | (3.970) | |
| **4 Coauthors** | | | -0.479*** | | -0.286*** |
| | | | (-5.393) | | (-3.186) |
| **5 Coauthors** | | | -0.726*** | | -0.487*** |
| | | | (-4.383) | | (-2.928) |
| **6 Coauthors** | | | -0.049 | | 0.161 |
| | | | (-0.164) | | (0.536) |
| **Time Fixed Effects** | Yes | Yes | Yes | Yes | Yes |
| **Observations** | 21,362 | 21,362 | 21,362 | 21,362 | 21,362 |
| **Log Likelihood** | -14,465.9 | -14,443.9 | -14,448.8 | -14,309.5 | -14,325.9 |
| **Akaike Inf. Crit.** | 28,973.8 | 28,933.9 | 28,943.6 | 28,667.0 | 28,699.8 |

*$p < .050$

**$p < .010$

***$p < .001$; $z$-stats are reported parenthetically below each coefficient.

The final analysis primarily assesses Hypothesis 4, but also serves to further assess Hypothesis 3. Hypothesis 4 is that journal article coauthoring has a greater impact on article quality than does alphabetical ordering of surnames, and as a reminder, Hypothesis 4 is that teamwork is positively associated with article quality. To assess these hypotheses, we predict article quality by alphabetical ordering and whether an article has coauthors, controlling for year fixed effects. As discussed, the dataset here includes papers written by a single author, whereas all previous analyses excluded these articles. Our findings are summarized in Table 5 below.

In treatment (1), we assess the impact that author alphabetical ordering has on article quality. Although this is a different dataset, the results are consistent with those outlined above, namely that alphabetical ordering is positively related to quality, indicated by a positive and statistically significant regression coefficient ($B_1 = 0.302$, $p < .001$). Treatment (3) indicates a similar finding for coauthored papers ($B_2 = -0.327$, $p < .001$), indicating that teamwork is associated with quality—with the opposite being shown to be the case in treatment (2). Further support is found for Hypothesis 3. Teamwork is positively associated with quality.

Treatment (4) includes the variables for alphabetical ordering and coauthoring, which both exhibit positive and statistically significant coefficients. However, the coauthoring coefficient ($B_2 = 0.534$, $p < .001$) is greater than that of alphabetical ordering ($B_1 = 0.468$, $p < .001$), indicating that coauthoring has greater impact than alphabetical ordering, albeit there is the potential that the confidence intervals could overlap. Hypothesis 4 is generally supported. Journal article coauthoring likely has a greater impact on article quality than does alphabetical ordering.

**Table 5. Binary logistic regression: Dependent variable is article quality (High = 1).**

|  | (1) | (2) | (3) | (4) |
|---|---|---|---|---|
| **Constant** | 0.110 | 0.342*** | 0.016 | -0.449*** |
|  | (1.924) | (6.218) | (0.257) | (-6.673) |
| **Alpha. Ordering** | 0.302*** |  |  | 0.468*** |
|  | (11.622) |  |  | (16.626) |
| **Single Authored** |  | -0.327*** |  |  |
|  |  | (-10.516) |  |  |
| **Coauthored** |  |  | 0.327*** | 0.534*** |
|  |  |  | (10.516) | (15.900) |
| **Year Fixed Effects** | Yes | Yes | Yes | Yes |
| **Observations** | 26,720 | 26,720 | 26,720 | 26,720 |
| **Log Likelihood** | -18,149.8 | -18,162.1 | -18,162.1 | -18,023.2 |
| **Akaike Inf. Crit.** | 36,341.6 | 36,366.2 | 36,366.2 | 36,090.5 |

*$p < .050$

**$p < .010$

***$p < .001$; $z$-stats are reported parenthetically below each coefficient.

## Discussion

### Theoretical and practical implications

Like many other disciplines [6, 13], team size among business scholars is increasing over time. However, we find that among business academics, alphabetical ordering disincentivizes author teamwork and results in fewer authors per publication than for contribution-based ordering of authors. Similar to studies in other academic disciplines [16–19], this research concludes that article authoring team size is related to article quality. Unlike prior studies, we further break down this relationship, finding that articles written by a single author are of lower quality than articles published by coauthors, but the number of coauthors exhibits decreasing returns to scale—coauthoring teams of one to three are positively related to high-quality articles, but larger teams are not. Alphabetical ordering itself, however, is *positively* associated with quality even though it inhibits teamwork, but journal article coauthoring has a greater impact on article quality than does alphabetical ordering.

These findings, taken together, suggest academic publishing policy changes—academic scholars respond to incentives [1, 2] and these incentives are currently misaligned. Specifically, to drive the best research, teamwork should be incentivized. Although some academic fields do order authors by contribution [5], this study provides a strong rationale that all fields should require coauthorship ordering based on author contribution levels and not by the spelling of author surname.

### Future research

This study infers that alphabetical ordering conventions in journal publication are detrimental to academic advancement because of inhibited teamwork, but only explicitly proves that alphabetical ordering is associated with smaller teams. We propose future research on the makeup of these smaller teams. For example, these smaller teams could be composed of more experienced researchers, as the penalty is high to add junior researchers when an article is ordered alphabetically [5]. We also propose future research to determine causality, with the contention that career progression in academia requires recognition, which is inhibited

by alphabetical ordering in journals, therefore incentivizing authors to publish in smaller groups.

## Conclusion

In academia, many institutions use journal article publication productivity for making decisions on tenure and promotion, funding grants, and rewarding stellar scholars. Although non-alphabetic sequencing of article coauthoring signals the extent to which a scholar has contributed to a project, many disciplines in academia follow the norm of alphabetical ordering of coauthors in journal publications. Generally, this study concludes that that author alphabetical ordering disincentivizes teamwork and reduces the overall quality of scholarship. Based on these findings we recommend that, to drive the highest-quality research, teamwork should be incentivized, and therefore all fields should adopt a contribution-based journal article author-ordering convention and avoid author ordering based upon the spelling of surnames. Although this study was undertaken using articles from business journals, its findings should generalize across all academia.

## Author Contributions

**Conceptualization:** Steven T. Joanis.

**Data curation:** Steven T. Joanis.

**Formal analysis:** Steven T. Joanis, Vivek H. Patil.

**Investigation:** Steven T. Joanis, Vivek H. Patil.

**Methodology:** Steven T. Joanis, Vivek H. Patil.

**Project administration:** Steven T. Joanis.

**Resources:** Steven T. Joanis.

**Software:** Steven T. Joanis, Vivek H. Patil.

**Supervision:** Vivek H. Patil.

**Validation:** Vivek H. Patil.

**Visualization:** Steven T. Joanis.

**Writing – original draft:** Steven T. Joanis.

**Writing – review & editing:** Steven T. Joanis, Vivek H. Patil.

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
