## [Decision Letter · Decision Letter 0]

22 Mar 2021

PONE-D-20-40522

Alphabetical ordering of author surnames in academic publishing: a detriment to teamwork

PLOS ONE

Dear Dr. Joanis,

Thank you for submitting your manuscript to PLOS ONE. After careful consideration, we feel that it has merit but does not fully meet PLOS ONE’s publication criteria as it currently stands. Therefore, we invite you to submit a revised version of the manuscript that addresses the points raised during the review process.

Your paper has been scientifically judged by an acknowledged expert in this field of knowledge. Overall, our referee recommends a major revision of your paper, on the basis of some key issues related to the study setting (i.e., purpose, hypotheses and their support and coherence, discussion, among others). Please responde to all these points in the rebuttal letter, properly detailing all the changes and amendments made in the revised paper.

We look forward to receiving your revised manuscript.

Kind regards,

Sergio A. Useche, Ph.D.

Academic Editor

PLOS ONE

Journal Requirements:

Reviewers' comments:

Reviewer's Responses to Questions

**Comments to the Author**

1. Is the manuscript technically sound, and do the data support the conclusions?

Reviewer #1: Partly

2. Has the statistical analysis been performed appropriately and rigorously? 

Reviewer #1: I Don't Know

3. Have the authors made all data underlying the findings in their manuscript fully available?

Reviewer #1: Yes

4. Is the manuscript presented in an intelligible fashion and written in standard English?

Reviewer #1: Yes

5. Review Comments to the Author

Reviewer #1: This paper describes an analysis of authorship based on alphabetical ordering and its impact on quality and number of coauthors, which the authors suggest is a proxy for teamwork. The paper is organized around five hypotheses, although the significance of hypotheses is always readily apparent from the literature cited. Generally, the structure of the manuscript is non-standard, and is confusing and repetitive. The research design and methodology are clearly described and well-written. The Results are overly long and redundant. The Discussion—named Summary in this submission—is a summary and does not clearly interpret the importance of the findings or place them in the context of existing literature.

There are a number of suggestions for improving the manuscript:

1. After the Purpose statement on p. 4, the authors provide an abbreviated presentation of the Methods and Results (lines 79-90), before listing the specific hypotheses guiding the study. This redundancy unnecessarily lengthens the paper and is confusing.

2. Why is Hypothesis 1 interesting or important? It states, “The continuum of alphabetical ordering of coauthorship will be anchored by accounting and marketing, with accounting having the greatest incidence of alphabetical ordering and marketing having the least. We anticipate that the incidence of alphabetization for organizational behavior and business technology lies between those two anchors.” There is no explanation of the importance of this comparison nor is its significance explained in the Discussion.

3. The authors somewhat consistently use the term “author ordering,” which is helpful to readers. However, in the background related to quality of scholarship, the authors use the term “authorship naming”… the introduction of a new term is not helpful for readers.

4. The authors write, “…the magnitude of the correlation between high rank and alphabetical ordering is more than four times that of the correlation between high rank and alphabetical ordering” (p. 15, line 262-263). This appears to be an error.

5. Perhaps another error, “the negative coefficients for the ger number of authors in

Treatment”, (p. 24, line 391).

6. Legends for the figures were not included in the manuscript.

6. PLOS authors have the option to publish the peer review history of their article (what does this mean?). If published, this will include your full peer review and any attached files.

Reviewer #1: No

---

## [Author Response · Author response to Decision Letter 0]

29 Mar 2021

Response to Reviewers

Article: Alphabetical ordering of author surnames in academic publishing: A detriment to teamwork

Steven T. Joanis

Department of Economics and Finance

Heider College of Business, Creighton University

602 North 20th Street

Omaha, NE 68102

Editorial Staff,

We very much appreciated the reviewer’s time in assessing this manuscript and found the comments exceptionally helpful in making this a better research study. Below are the corrective actions that we have undertaken to prepare this article for publication in PLOS One. 

Point 1. “Please ensure that your manuscript meets PLOS ONE's style requirements, including those for file naming.”

We followed the style guides provided by PLOS One for all aspects of the submission (main body, title/authors/affiliations, etc.).

We also uploaded the figure files to the Preflight Analysis and Conversion Engine digital diagnostic tool, to ensure that figures meet PLOS requirements.

Point 2. “We note that you have indicated that data from this study are available upon request. PLOS only allows data to be available upon request if there are legal or ethical restrictions on sharing data publicly.”

We uploaded all data required to replicate all analyses conducted in the writing of this manuscript.

The data depository used is: DANS (Data Archiving and Networked Services). The DOI is: https://doi.org/10.17026/dans-z72-4qp3

Point 3: “Is the manuscript technically sound, and do the data support the conclusions? The manuscript must describe a technically sound piece of scientific research with data that supports the conclusions. Experiments must have been conducted rigorously, with appropriate controls, replication, and sample sizes. The conclusions must be drawn appropriately based on the data presented. Reviewer #1: Partly.”

We believe that this reviewer comment was driven by our “non-standard” drafting of the original manuscript, as discussed by the reviewer in Point 6 below.

To address this criticism, we changed the format of the paper to mirror the generalized organization/structure observed during a comprehensive review that we undertook of the 2021 published research papers by PLOS One. Our approach is discussed further in Point 6 below. 

In addition, we changed our previously titled “Summary” Section to “Discussion” and added a clear and concise “Conclusion” section.

Point 4: “Has the statistical analysis been performed appropriately and rigorously? Reviewer #1: I Don't Know.”

To ensure correctness of all analyses and tables, we reran all statistical analyses to ensure the work was correctly performed and checked all resultant tables for accuracy.

We found no mistakes or anomalies.

Point 5: “Have the authors made all data underlying the findings in their manuscript fully available? The PLOS Data policy requires authors to make all data underlying the findings described in their manuscript fully available without restriction, with rare exception. The data should be provided as part of the manuscript or its supporting information or deposited to a public repository. Reviewer #1: Yes.”

Please refer to Point 2 above. We have provided this data as requested. 

Point 6: “This paper describes an analysis of authorship based on alphabetical ordering and its impact on quality and number of coauthors, which the authors suggest is a proxy for teamwork. The paper is organized around five hypotheses, although the significance of hypotheses is always readily apparent from the literature cited. Generally, the structure of the manuscript is non-standard, and is confusing and repetitive.”

As discussed in Point 3 above, this reviewer comment prompted us to undertake a comprehensive review of all the manuscripts published by PLOS One in 2021. We specifically focused on studies that used regression as the primary statistical technique. 

Although there is not a specific article structure used by all manuscripts, we identified a structure that was the most similar to most articles that had been published recently in PLOS One, and we reorganized the structure of our paper to mirror this. 

Point 7: “The Results are overly long and redundant.” 

We streamlined the Results Section by removing redundant information.

Overall, the Results Section had a word count reduction of 19.1%. 

Point 8: “The Discussion—named Summary in this submission—is a summary and does not clearly interpret the importance of the findings or place them in the context of existing literature.” 

As mentioned in Point 3 above, we split the previously titled “Summary” section into sections titled “Discussion” and “Conclusion”.

In addition, we took the advice of specifically discussing the interpretation of the results and their importance and placed them in the context of existing studies, to demonstrate how this study advances the literature. 

Point 9: “After the Purpose statement on p. 4, the authors provide an abbreviated presentation of the Methods and Results (lines 79-90), before listing the specific hypotheses guiding the study. This redundancy unnecessarily lengthens the paper and is confusing.” 

We removed this information from this section and incorporated it into the Methods section of the manuscript.

Point 10: “Why is Hypothesis 1 interesting or important? It states, “The continuum of alphabetical ordering of coauthorship will be anchored by accounting and marketing, with accounting having the greatest incidence of alphabetical ordering and marketing having the least. We anticipate that the incidence of alphabetization for organizational behavior and business technology lies between those two anchors.” There is no explanation of the importance of this comparison nor is its significance explained in the Discussion.”

We found this to be the most important criticism of this manuscript. We also agree that the differences between the authoring conventions of the various business subdisciplines does not change the conclusions or recommendations of the paper, and we therefore removed this hypothesis and all aspects of the paper that referred to it.

We believe that the resultant revision of the paper removes a distraction that could have been confusing to the reader, and improves overall readability, making the main arguments of the paper stronger. 

Point 11: “The authors somewhat consistently use the term “author ordering,” which is helpful to readers. However, in the background related to quality of scholarship, the authors use the term “authorship naming”. The introduction of a new term is not helpful for readers.”

We modified this language, and all the terms referring to “author ordering” throughout the manuscript, for consistency and clarity. 

Point 12: “The authors write, “…the magnitude of the correlation between high rank and alphabetical ordering is more than four times that of the correlation between high rank and alphabetical ordering” (p. 15, line 262-263). This appears to be an error. Perhaps another error, “the negative coefficients for the ger number of authors in Treatment”, (p. 24, line 391).”

We corrected these errors.

Point 13: “Legends for the figures were not included in the manuscript.” 

We ensured that legends and titles of all figures were appropriately included.

We are grateful for the comments on our manuscript and believe that they have made this paper much better than it was previously. 

We thank you for your consideration and look forward to your comments on this revised manuscript. 

Regards,

Steven T. Joanis

---

## [Decision Letter · Decision Letter 1]

22 Apr 2021

Alphabetical ordering of author surnames in academic publishing: a detriment to teamwork

PONE-D-20-40522R1

Dear Dr. Joanis,

We’re pleased to inform you that your manuscript has been judged scientifically suitable for publication and will be formally accepted for publication once it meets all outstanding technical requirements.

Kind regards,

Sergio A. Useche, Ph.D.

Academic Editor

PLOS ONE

Additional Editor Comments (optional):

Reviewers' comments:

Reviewer's Responses to Questions

**Comments to the Author**

1. If the authors have adequately addressed your comments raised in a previous round of review and you feel that this manuscript is now acceptable for publication, you may indicate that here to bypass the “Comments to the Author” section, enter your conflict of interest statement in the “Confidential to Editor” section, and submit your "Accept" recommendation.

Reviewer #1: All comments have been addressed

2. Is the manuscript technically sound, and do the data support the conclusions?

Reviewer #1: Yes

3. Has the statistical analysis been performed appropriately and rigorously? 

Reviewer #1: Yes

4. Have the authors made all data underlying the findings in their manuscript fully available?

Reviewer #1: Yes

5. Is the manuscript presented in an intelligible fashion and written in standard English?

Reviewer #1: Yes

6. Review Comments to the Author

Reviewer #1: The manuscript is greatly improved in terms of clarity and organization. The authors have been thoughtful in response to earlier review comments.

7. PLOS authors have the option to publish the peer review history of their article (what does this mean?). If published, this will include your full peer review and any attached files.

Reviewer #1: No

---

## [Editor Report · Acceptance letter]

27 Apr 2021

PONE-D-20-40522R1 

Alphabetical ordering of author surnames in academic publishing: a detriment to teamwork 

Dear Dr. Joanis:

I'm pleased to inform you that your manuscript has been deemed suitable for publication in PLOS ONE. Congratulations! Your manuscript is now with our production department. 

Kind regards, 

on behalf of

Dr. Sergio A. Useche 

Academic Editor

PLOS ONE